# Effects of *Gnaphalium affine* Extract on the Gel Properties of •OH-Induced Oxidation of Myofibrillar Proteins

**DOI:** 10.3390/foods13101447

**Published:** 2024-05-08

**Authors:** Haijun Chang, Yu Hu, Yuanwei Shi, Jie Xiong, Zhaoying Bo

**Affiliations:** Chongqing Engineering Research Center for Processing, Storage and Transportation of Characterized Agro-Products, College of Environment and Resources, Chongqing Technology and Business University, No.19 Xuefu Ave., Nan’an District, Chongqing 400067, China; huyu176@163.com (Y.H.); shiyuanwei1212@163.com (Y.S.); luckyjier3@163.com (J.X.); zhao15736366466@163.com (Z.B.)

**Keywords:** *Gnaphalium affine* extract (GAE), tea polyphenols (TP), myofibrillar proteins (MPs), oxidation, gel properties

## Abstract

This study aimed to investigate the effect of *Gnaphalium affine* extract (GAE) (0.04, 0.2 and 1 mg/g protein) on the gel properties of porcine myofibrillar proteins (MPs) in a simulated Fenton oxidation system, using tea polyphenols (TPs) at similar concentrations of 0.04, 0.2, and 1 mg/g protein, respectively, as a contrast. The findings revealed that as the TP concentration increased, the water retention of MP gels decreased significantly (*p* < 0.05). In contrast, MP gels containing medium and high concentrations of GAE exhibited significantly higher water retention than those with low concentrations of GAE (*p* < 0.05). When the concentration of GAE was increased to 1 mg/g protein, the strength of MP gels was significantly reduced (*p* < 0.05) by 33.32% compared with the oxidized control group, suggesting that low and medium GAE concentrations support MP gel formation. A texture profile analysis indicated that an appropriate GAE concentration improved gel structure and texture. Dynamic rheological characterization revealed that low concentrations of TP (0.04 mg/g protein) and low and medium concentrations of GAE (0.04 and 0.2 mg/g protein) strengthened the protein gel system. Conversely, high concentrations of TP and GAE (1.0 mg/g protein) damaged the protein gel system or even promoted the collapse of the gel system. Scanning electron microscopy revealed that higher TP concentrations disrupted the gel, whereas low and medium GAE concentrations maintained a more continuous and complete gel network structure compared with the oxidized control group. This indicates that an appropriate GAE concentration could effectively hinder the destruction of the gel network structure by oxidation. Therefore, based on the obtained results, 0.2 mg/g protein is recommended as the ideal concentration of GAE to be used in actual meat processing to regulate the oxidization and gel properties of meat products.

## 1. Introduction

Myofibrillar proteins (MPs) comprise approximately 50~60% of the total protein in meat and possess various functional properties (water binding, gelation, emulsification, etc.) [1]. Gelatinized meat products are favored by a wide range of consumers for their smoothness, elasticity, and high nutritional properties. Therefore, enhancing the gelatinization properties of minced meat products is crucial for producing higher-quality meat products [2]. However, meat products are vulnerable to oxidation due to exposure to oxygen in the air and catalytic actions of iron, copper, and other metal ions in metal instruments during processing, transportation, sale, and storage. This oxidation can cause discoloration, off-flavors, deterioration of texture, and loss of nutrients [3].

With the increased health awareness among people, modern consumers prefer natural antioxidants over synthetic ones (such as butylated hydroxyanisole, 2,6-di-*tert*-butyl-4-methylphenol, and *tert*-butylhydroquinone), which have safety concerns and potential toxicity. Plant polyphenol extracts, for example, green tea extract [4], clove extract [5], etc., are widely used as natural antioxidants to prevent meat oxidation during processing, thereby improving the taste, flavor, and quality of meat products. Using natural antioxidants instead of synthetic antioxidants in food processing has become the trend. 

Plant polyphenols are considered the most effective natural antioxidants in recent years. Recent studies have focused more on controlling protein oxidation by adding polyphenols and exploring how these additives influence the properties of myofibrillar fibrous protein gels [6,7]. The main reason is that the interaction between polyphenols and myofibrillar fibrous protein affects the amino acid side-chains of the protein, inducing changes in its secondary and tertiary conformations, which in turn alter its functional properties, especially its gelation properties [8]. *Gnaphalium affine*, a traditional wild vegetable commonly featured during the Qingming festival held in China, is rich in polyphenols and flavonoids and exhibits antibacterial, anti-inflammatory, and antioxidant effects [9]. Existing studies have shown that *G. affine* extract (GAE) has a strong free radical scavenging ability, especially against •OH, and can effectively inhibit lipid oxidation, suggesting its potential as an antioxidant [10,11].

Despite these properties, the impact of GAE on protein oxidation and the structural and functional properties of proteins requires further exploration. Our previous research found that GAE was rich in flavonoids and phenolic compounds with strong scavenging abilities against free radicals, especially •OH [12]. Therefore, investigating the effect of GAE on the gel properties of MPs in the context of its ability to resist MP oxidation is of great significance for meat processing. 

The regulation of the gel properties of MPs with GAE in a simulated Fenton oxidation system has not been reported to date. Therefore, further studies are needed to determine whether GAE can influence the gel properties of MPs under oxidizing conditions. This study aimed to investigate the effect of GAE on the gel properties of porcine MPs in a simulated Fenton oxidation system, using tea polyphenols (TP) as a control. Heat-induced gels were first prepared by adding different concentrations of TP/GAE to a simulated Fenton oxidation system. The relevant indexes of gel properties (i.e., gel strength, gel texture, gel water retention, gel cooking yield, gel whiteness, rheological properties, and gel microstructure) were determined to investigate the effects of TP and GAE on the gel properties of MPs. This study provides a practical reference for the scientific application of natural plant extracts rich in phenols in the “green” processing of meat products. 

## 2. Materials and Methods

### 2.1. Materials 

Fresh pork *Longissimus dorsi* muscle was purchased from the local Renrenle supermarket (Nanan, Chongqing, China). All visible fat and connective tissue were removed, and the meat was cut into uniform pieces (approximately 50 g each), packed in vacuum bags (DZ-500/2S, Zhucheng Kuntai Food Machinery Co., Ltd., Zhucheng, China), and then stored at −20 °C (BCD-215KHN, Qingdao Haier Co., Ltd., Qingdao, China) until further use. Fresh *Gnaphalium affine* was sourced from Linshui County, located in Guangan City, Sichuan Province of China, coinciding with the Qingming festival. It was naturally dehydrated, then sealed and stored in a desiccator for future use. Tea polyphenols (TPs) (97%), L-ascorbic acid, bovine serum albumin (BSA), and other chemicals of analytical grade were obtained from Aladdin Biochemical Technology Co., Ltd. (Shanghai, China). 

### 2.2. Gnaphalium affine Extract (GAE) Preparation

*Gnaphalium affine* was dried in a DGG-9076A electric oven (Shanghai Qixin Scientific Instrument Co., Ltd., Shanghai, China) at 60 °C for 4 h and then crushed and filtered through a 100-mesh sieve to obtain the dried powder. GAE was prepared by mixing 50 g of the dried *Gnaphalium affine* powder with 1000 mL of ethanol solution (70%, *v*/*v*). The mixture was extracted for 80 min at 45 °C using an ultrasonic processor (KQ-1500DE, Kunshan Ultrasound Equipment Co., Ltd., Kunshan, China) at 40 kHz and 1500 W. The extract was vacuum-filtered, and the filtrate was concentrated under a vacuum at 60 °C to 100 mL. This extract was then freeze-dried (Alpha 2-1.2, Christ, Osterode, Germany) for 48 h under a cold trap at −30 °C and a vacuum of −20 Pa until the moisture content reached approximately 5%. It was then sealed and stored in a brown, dry dish for later use. 

### 2.3. Extraction of MPs and Oxidative Treatments with TP/GAE

#### 2.3.1. Extraction of MPs

MPs were extracted from the pork *Longissimus dorsi* muscle following the method described by Park et al. [13], with some modifications. Pork (50 g) was thawed at 4 °C for 4 h and then homogenized and washed with five volumes of stiffening buffer (10 mmol/L sodium phosphate, 0.1 mol/L NaCl, 2 mmol/L MgCl_2_, and 1 mmol/L EGTA; pH 7.0). The resulting mixture was dispersed for 1 min using a high-speed disperser (Ultra-Turrax T25, IKA-WERKE, Staufen im Breisgau, Germany) and centrifuged (6000× *g*, 10 min, 4 °C) (TGL-20 High speed refrigerated centrifuge, Sichuan Shuke Instrument Co., Ltd., Chengdu, China). The supernatant was decanted, and the precipitate was resuspended in five volumes of stiffening buffer; this was repeated three times. The precipitate was then homogenized in five volumes of 0.1 mol/L NaCl solution and filtered to remove residual connective tissue. The pH was adjusted to 7.0 using 0.1 mol/L HCl. The resulting white, paste-like precipitate was identified as the MPs and was stored on crushed ice to be used within 48 h. The protein concentration was determined by the biuret method using bovine serum albumin (BSA) as the standard.

#### 2.3.2. Oxidative Treatments with TP/GAE

The MP paste was diluted to 40 mg/mL with an oxidation system solution (30 μmol/L FeCl_3_, 100 μmol/L ascorbic acid, and 3 mmol/L H_2_O_2_, dissolved in 15 mmol/L PIPES buffer at pH 6.25) after adding different volumes of TP (5 mg/mL, dissolved in oxidizing solution) and GAE solutions (5 mg/mL, dissolved in oxidizing solution), and left for 12 h at 4 °C. The oxidation reaction was then terminated by adding Trolox (1 mmol/L, final concentration). The blank group and the oxidation control group were set up concurrently, and the experimental samples are shown in Table 1.

### 2.4. Preparation and Characterization of Heat-Induced Gels

#### 2.4.1. Preparation of Heat-Induced Gels

Heat-induced gels were prepared according to a modified previous study [14] with slight modifications. The MP samples of terminated oxidation were heated at room temperature up to 80 °C and held for 10 min in a constant-temperature water bath (Guohua HH-42, Changzhou Guohua Electric Co., Ltd., Changzhou, China), cooled in an ice bath, and stored overnight at 4 °C. The gel samples were then brought to room temperature for 1 h before gel properties were determined.

#### 2.4.2. Cooking Yield and Water Retention of MP Gels

The total weight of the centrifuge tube and the MPs was accurately measured before cooking (*W*_1_), and after the water was removed post-heating (*W*_2_). The cooking yield of the MP gel was calculated according to Equation (1).
(1)$$\mathrm{Cooking}\;\mathrm{yield}\;\mathrm{of}\;\mathrm{gel}\;\%=\frac{W_{2}}{W_{1}}\times 100\%$$

The water-holding capacity (WHC) of the MP gel was determined using a modified method from Ma et al. [15]. The gel was weighed after removing the water by centrifugation (4 °C, 10,000 rpm, 10 min). The water retention was calculated according to Equation (2), where *W*_1_ is the total weight (g) of the centrifuge tube and gel after centrifugal dewatering; *W*_2_ is the total weight (g) of the centrifuge tube and gel before centrifugation; and *W* is the weight (g) of the empty centrifuge tube.
(2)$$\mathrm{WHC}\;\%=\frac{W_{1}-\;W}{W_{2\;}-\;W}\;\times \;100\%$$

#### 2.4.3. Whiteness of MP Gels

The chromatic aberration of the MP gel was measured using a Minolta Chroma Meter CR-400 (Konica Minolta Sensing, Inc., Tokyo, Japan). The samples were examined after self-check, zero point, and whiteboard correction. The gel whiteness was calculated according to Equation (3), where *L** is the luminance value, *a** is the red value, and *b** is the yellowness value.
(3)$$\mathrm{Gel}\;\mathrm{whiteness}=100\;-\sqrt{{\left(100-L*\right)}^{2}{+(a*)}^{2}{+(b*)}^{2}}$$

#### 2.4.4. Strength and Texture of MP Gels

The strength and texture of the gels were measured using a TA-XT2i texture analyzer (Stable Micro Systems Ltd., Godalming, UK) and the Texture Expert for Windows software (Version 1.0, Stable Micro Systems Ltd.) as described in Chang et al. [16,17] with modifications. The gel strength was measured at room temperature using a P/0.5 piston with the following parameters: pretest speed, 2.00 mm/s; test speed, 1.00 mm/s; post-test speed, 2.00 mm/s; penetration distance, 6 mm; trigger force, 5.0 g; and data acquisition rate, 200 points per second (PPS). 

The texture profile analysis (TPA) of the samples was conducted at room temperature using a P/75 piston with the following parameters: pretest speed, 2.00 mm/s; test speed, 1.00 mm/s; post-test speed, 1.00 mm/s; compression ratio, 50% with a rest period of 5 s between two cycles; trigger force, 5.0 g; and data acquisition rate, 200 PPS. The probe always returned to the trigger point before starting the second cycle. After the second cycle, the probe returned to its initial position. The following six TPA parameters (hardness, springiness, cohesiveness, gumminess, chewiness, and resilience) were calculated using the User Guide Software (Version 1.0, Stable Micro Systems Ltd.). The data obtained from the TPA curve were used for calculating textural parameters. 

#### 2.4.5. Dynamic Rheology of MP Gels

The MP samples with different treatments were centrifugally degassed (4 °C, 1000 rpm, 30 s), and then the rheological storage modulus (*G*′) and loss modulus (*G*″) were measured using a rotary rheometer (HAAKE MARS40, Thermo Fisher Scientific Technology Co., Ltd., Waltham, MA, USA). The P20/TI-01210465 rotor was installed in oscillatory temperature scanning test mode; the oscillation frequency was 0.1 Hz, the maximum stress was 2%, and the upper and lower plate slits were 1 mm. The samples were heated at a rate of 2.1 °C/min, and the heating curves at 20 °C~80 °C were recorded. Silicone oil was applied to the edges after scraping to prevent evaporation of the protein solution during the heating process. 

#### 2.4.6. Microstructure of MP Gels

The microstructure of MP gels was analyzed using a scanning electron microscope (SEM) after preparing the samples following the procedures reported in Jia et al. [18] with some modifications. The gel pieces (2 × 2 × 2 mm^3^) were fixed for 24 h in 10 mL glutaraldehyde solution [2.5% (*v*/*v*) mixed with 10 mmol/L phosphate-buffered saline (PBS), pH 6.8], and then rinsed with PBS (0.1 mol/L, pH 6.8) three times for 10 min each. Subsequently, the samples were dehydrated in different ethanol concentrations (50%, 70%, 80%, and 90%) for 10 min at each concentration at room temperature and then in absolute ethanol three times for 20 min each. The samples were treated with 100% ethanol and *tert*-butanol (1:1) three times for 30 min each. Following freeze-drying and sputter coating with a gold film (10 nm), the specimens were observed using an SEM at 2000× magnification (S-8020, Hitachi High-Technologies Corporation, Tokyo, Japan) and an accelerating voltage of 15.0 kV.

### 2.5. Statistical Analysis

The experiments were performed in triplicate and the data are expressed as the average of three trials (mean ± standard deviation). Statistical analyses were carried out using SPSS 19.0 (SPSS Inc., Chicago, IL, USA). A one-way analysis of variance (ANOVA) and Duncan’s multiple-range test were performed to determine the differences between the samples from the various groups at a significance level of *p* < 0.05.

## 3. Results and Discussion

### 3.1. Cooking Yield of MP Gels

As shown in Figure 1, the cooking yield of MP gels in the oxidized group decreased significantly by 16.98% compared with that in the non-Fenton oxidizing system (*p* < 0.05). Oxidation compromised the protein structure, leading to a reduced cooking rate of MP gels. This reduction impaired the water retention capacity of the gels’ spatial structure, resulting in partial water precipitation [19]. No significant differences in the cooking yields were observed between the two groups when 0.04 and 0.2 mg/g protein of TP were added compared with the oxidation control group (*p* > 0.05). However, when the concentration of TP was increased to 1 mg/g protein, the cooking yield of MP gels decreased significantly by 46.86% (*p* < 0.05). This suggests that a high concentration of TP further damaged the spatial structure of the MPs and reduced their ability to retain water molecules. Additionally, the addition of low, medium, and high concentrations of GAE did not significantly affect the cooking yields of MP gels compared with the oxidized control group (*p* > 0.05), suggesting that GAE had a minimal effect on the spatial structure of MP gels and did not contribute to additional water loss.

### 3.2. Water Retention of MP Gels

The MP gels, thermally induced and cooled, formed a dense three-dimensional network structure that effectively locked in most of the water. Therefore, the water retention of the gel reflects the capacity of the protein gel network. As depicted in Figure 2, the water retention of the MP gels in the oxidized control group showed a slight decrease compared with the unoxidized group, although this difference was not statistically significant (*p* > 0.05). Notably, the water retention significantly decreased by 26.77% with increasing concentrations of TP, particularly when the TP concentration reached 1 mg/g protein, compared with the oxidized control group (*p* < 0.05). Low concentrations of GAE seemed to exacerbate the oxidative degradation of gel properties, resulting in a significant decrease in water retention (*p* < 0.05). However, the water retention of MP gels with medium and high concentrations of GAE was significantly higher than that with low concentrations (*p* < 0.05) and not significantly different from oxidized controls (*p* > 0.05). This indicates that medium and high concentrations of GAE did not compromise the water retention of the gels, likely due to the phenolics in GAE, which strongly bound free radicals, protected protein side-chain structures, and enhanced protein cross-linking, thus forming a more robust network with improved WHC [20,21]. A correlation was observed between the changes in water retention and the strength of MP gels in the TP group. TP interacted with proteins; the higher the concentration of TP, the more serious the damage to the amino acid side-chain structure of proteins and the rupture of the three-dimensional gel network of thermally induced MP gels, leading to a decrease in the water retention and strength of gels. 

### 3.3. Whiteness of MP Gels

Whiteness is an essential indicator of meat quality, intuitively reflecting the color and perceived freshness of the meat [22]. Typically, a higher whiteness value indicates a brighter color and suggests better quality, making it an essential measure of gel quality. As illustrated in Figure 3, no significant difference in the whiteness of oxidized MP gels was observed compared with the unoxidized group (*p* > 0.05), indicating that protein oxidation did not affect gel whiteness. However, the addition of TP/GAE significantly altered the whiteness (*p* < 0.05). Specifically, the whiteness of MP gels decreased progressively with increases in TP concentration, showing reductions of 3.47, 5.31, and 14.74 in the low-, medium-, and high-concentration groups, respectively (*p* < 0.05). No significant difference in the whiteness value was observed between the low-concentration GAE group and the oxidation control group (*p* > 0.05), but significant reductions of 2.83 and 7.05 were observed in the medium- and high-concentration GAE groups, respectively (*p* < 0.05). These decreases in whiteness value might be attributed to the interaction of TP/GAE with protein, which probably affected the protein’s spatial structure, loosened the gel network, reduced the free water content between gel tissues, weakened light scattering on the gel surface, and decreased the measured luminance *L** value, resulting in a lower whiteness value. Additionally, the decrease in the gel whiteness value might relate to the color of the oxidized TP/GAE, with the reddish-brown color of TP and the green color of GAE influencing the gel’s redness value (*a**) and yellowness value (*b**), respectively, which in turn affected the whiteness value.

These findings align with those of Cao and Xiong [21], who found that low concentrations of chlorogenic acid did not alter the whiteness values of MP gels, whereas medium and high concentrations significantly reduced them (*p* < 0.05). The chelation of chlorogenic acid with iron ions and the formation of darker quinones upon oxidation contributed to the reduced gel whiteness. Similarly, Ma et al. [5] reported that inulin treatment altered the color of MP gels, further decreasing their whiteness.

### 3.4. Strength of MP Gels

Gel strength, defined as the initial pressure required to puncture the gel, reflects the protein’s ability to form a robust gel [21]. This measure is an essential index for evaluating the properties of MP gels [23]. As demonstrated in Figure 4, the strength of MP gels in the oxidized control group was slightly lower compared with the unoxidized group, although the values were not significantly different (*p* > 0.05). Oxidation, which introduces free radicals, alters the protein structure and disrupts the stable three-dimensional network of MP gels, weakening their gel-forming capability [24]. The strength of MP gels in the oxidized group did not exhibit a significant decreasing trend, possibly due to the insufficient oxidizing intensity. The gel strength gradually reduced with increasing TP concentration; particularly, it diminished by 60.73% in the high-concentration TP group compared with the oxidized control group. As the concentration of GAE increased, the strength of MP gels appeared to change irregularly. It slightly increased at low and medium concentrations, though these changes were not statistically significant (*p* > 0.05). However, a significant reduction in strength, by 33.32%, occurred when the GAE concentration reached 1 mg/g of protein (*p* < 0.05), compared with the oxidized control group. This significant weakening could be attributed to the oxidation of TP/GAE leading to the formation of quinone derivatives, which, along with protein sulfhydryl groups or free amino groups, produced “sulfhydryl-quinone”/”amino-quinone” adducts. These compounds blocked the disulfide bonds crucial for gel formation during heating, causing the protein gel network to loosen and reducing the strength of the gels [14,25]. High concentrations of EGCG [26], catechins [27], and chlorogenic acid [21] have been shown to markedly decrease the strength of MP gels. In contrast, low and medium concentrations of GAE enhance gel formation, likely due to the quercetin content in GAE, as supported by Zhang et al. [28]. Higher concentrations of quercetin correlate with increased gel strength, suggesting that quercetin is beneficial for protein gel formation [28].

### 3.5. Texture of MP Gels

Gel hardness is a key indicator of a gel’s textural properties, indicating the maximum peak pressure achieved by the instrument probe during its initial compression of the gel. As depicted in Table 2, MP gels that underwent oxidation and subsequent addition of TP/GAE exhibited significantly reduced hardness compared with the unoxidized group (*p* < 0.05). Additionally, the gel hardness gradually declined with the increase in TP concentration, indicating that TP disrupted the formation of the network structure of MP gels, thereby softening them. Conversely, after adding GAE, gel hardness initially increased and then decreased as the GAE concentration gradually increased. Notably, the medium-concentration GAE group (0.2 mg/g protein) displayed the highest gel hardness, which was significantly greater than that in the oxidized control group (*p* < 0.05). This suggests that moderate GAE concentrations effectively protected protein gels from oxidative damage, whereas high concentrations impaired gel properties, similar to the effects of TP. This was probably because excessive TP/GAE occupied the interior of the protein molecules and prevented cross-linking between them, leading to structural breaks in the gel network and deterioration of gel hardness and other properties.

Gel elasticity refers to the gel’s ability to rebound after initial compression by the probe, reflecting its capacity to deform under external force and recover after the force is removed [29]. The elasticity of oxidized MP gels was significantly lower than that of the unoxidized group (*p* < 0.05). In the presence of increasing TP concentrations, elasticity generally decreased, with the exception of the low-concentration group, which showed a slight increase followed by a gradual decrease. In the presence of GAE, the gel elasticity was slightly higher in the 0.2-mg/g protein GAE group than in the oxidized control group, while both the low- and high-concentration GAE groups showed a marked decrease.

Cohesion refers to the internal bonding force that produces the desired form of a food product. Resilience is the degree to which a food product returns to its original shape after deformation, under the same conditions of speed and pressure that caused the deformation. The trend of cohesion in each group was consistent with the trend of elasticity. Oxidation notably reduced the resilience of MP gels (*p* < 0.05), with higher TP concentrations correlating with diminished gel resilience. The resilience in the GAE-treated group did not significantly differ from that of the oxidized control group (*p* > 0.05) [24,30]. The trends of gel viscosity and chewability in all groups were consistent with those of elasticity and cohesion, highlighting the destructive impact of oxidation on the texture of MP gels, which resulted in a significant decline in gel hardness, elasticity, cohesion, adhesiveness, chewability, and resilience (*p* < 0.05). The destructive force gradually increased at medium and high TP concentrations; however, a low TP concentration slightly modified the gel texture. Thus, an appropriate concentration of GAE modified the gel structure and improved its texture.

### 3.6. Dynamic Rheometry of MP Gels

Dynamic oscillatory rheological characterization is commonly used to characterize the viscoelastic properties of protein gels. In this technique, the energy storage modulus (*G*′), which represents the amount of recoverable energy stored in an elastic gel, is plotted against temperature. This plot serves as a critical tool for understanding how the elasticity of a protein gel system changes with increasing temperature [18].

According to Figure 5, unoxidized MP samples showed an initial increase in the *G*′ value when heated to 50 °C. The first transition peak appeared at 62 °C, where the *G*′ value decreased rapidly, followed by a second peak at 73 °C, where the *G*′ value showed an increasing trend again. This pattern indicates that MP proteins gradually denatured and aggregated to form an elastic gel network during the heating process. In contrast, oxidized MP samples also showed two characteristic transition peaks, but these were lower than those in the unoxidized samples, and the *G*′ values at the end of heating were significantly reduced. This suggests that oxidation slightly disturbed the structural stability of proteins, damaging the gel network structure and reducing the elasticity of the resulting gel.

Compared with the oxidized control, the presence of a low concentration (0.04 mg/g protein) of TP significantly enhanced the two transformation peaks and contributed to higher final *G*′ values, which were even higher than those of the unoxidized group. This revealed that low concentrations of TP under oxidized conditions were beneficial for protein intermolecular interactions and cross-linking aggregation, facilitating the formation of a denser gel network structure during the heating process. This effect is attributed to the quinone produced by TP oxidation, which not only promoted the conversion of protein sulfhydryl groups into disulfide bonds, but also served as a cross-linking agent, covalently binding nucleophilic groups from different protein peptide chains to promote protein cross-linking. Additionally, the unique phenolic structure of TP could inhibit the oxidative destruction of protein structure, leading to a more stable protein solution system. Consequently, the heat-induced gel formed was of better quality and had higher elasticity. However, when TP was present at a medium concentration (0.2 mg/g protein), both transformation peaks and *G*′ values at the heating endpoint were significantly lower than those observed with low concentrations of TP, suggesting that this concentration adversely affected the protein conformation of the MPs and reduced the elasticity of the protein gel. This reduction was likely due to the extensive shielding by TP of the reactive functional groups, which prevented the aggregation of inter-protein cross-links and led to a lower-quality gel. The presence of a high concentration of TP (1.0 mg/g protein) exacerbated these effects, with the two typical transition peaks disappearing and severely damaging the gel properties of MPs. This severe disruption was caused by excess TP, which greatly disrupted the structure of the MPs and led to excessive aggregation, thus hindering the formation of the gel network during the heating process.

GAE also affected the elastic modulus of MP gels. For example, the transformation peak values and the *G*′ values at the heating endpoint in the low- and medium-concentration GAE groups (0.04 and 0.2 mg/g protein, respectively) increased significantly with increasing GAE concentration and were substantially higher than those in the unoxidized group. This indicates that GAE addition in this concentration range was beneficial for the formation of protein gels. However, when the GAE concentration was increased to 1.0 mg/g protein, both the characteristic transition peaks and the final G′ values were generally lower than those in the oxidized control group. This indicates that high concentrations of GAE still allowed for gel formation, despite the reduction in typical transition peaks and thermal induction. However, adding a low amount of GAE could effectively promote protein cross-linking and aggregation, forming a more robust gel network system due to the large amount of phenolics in GAE. Conversely, an excessive amount of GAE might lead to excessive aggregation of proteins or even shield the functional groups involved in the reaction, thereby compromising the stability of the protein structure and consequently deteriorating the elasticity of thermally induced gels.

The trend of the loss modulus (*G*″) was consistent with that of the energy storage modulus (*G*′), and the *G*″ value showed the same initial increase when the temperature reached 50 °C. The first transition peak occurred when the temperature increased to 62 °C, followed by a rapid decrease in the *G*″ value, which then remained constant. The *G*″ value remained lower than the *G*′ value during the gel formation process at increased temperatures, thus indicating that the elasticity of the protein gel system was much higher than the viscosity, but the two properties were positively correlated.

In summary, low concentrations of TP (0.04 mg/g protein) and low to medium concentrations of GAE (0.04 and 0.2 mg/g protein, respectively) strengthened the protein gel system. Conversely, high concentrations of TP and GAE (1.0 mg/g protein) disrupted the protein gel system or even led to its collapse, aligning with findings from numerous studies. Cao and Xiong [21] reported that low and medium concentrations of chlorogenic acid (6 and 30 μmol/g, respectively) promoted the *G*′ values of MPs, especially at the heating endpoint, while high concentrations (150 μmol/g) eliminated the two characteristic transition peaks, significantly impairing gel properties. Similarly, Jia et al. [18] observed that the trends of the *G*″ curve were consistent with those of the G′ curve, with elasticity playing a pivotal role. The addition of 10 μmol/g catechin did not change the trend of the *G*′ curve, and the *G*′ value at the endpoint of heating was higher than that of the blank control, enhancing the interaction between protein molecules. However, the addition of high concentrations of catechin (50, 100, and 200 μmol/g) resulted in excessive aggregation of MPs and a loss of gel properties. Feng et al. [26] reported that the addition of both 100 ppm and 1000 ppm EGCG significantly promoted the G′ value, thereby facilitating gel formation. This effect was likely due to EGCG’s interaction with proteins increasing their surface hydrophobicity, which in turn promoted gel contraction during heating, resulting in denser gels.

### 3.7. Microstructure of MP Gels

The gel microstructure, which includes the three-dimensional network structure, pore size, and degree of densification, could be visualized using an SEM. These characteristics were crucial for evaluating the functional properties of the gel [31]. As illustrated in Figure 6, unoxidized MP gels showed a continuous protein network structure. However, this distinct network structure disappeared after oxidation, showing large flakes. This change was likely due to oxidation promoting the cross-linking and aggregation of some proteins while simultaneously destroying the network structure [26]. With the gradual increase in the TP concentration, the MP gels began to fragment, leading to a loosening of its structure and an increase in the number of gel pores. This indicates that TP interfered with the formation of MP gels, with higher concentrations proving more destructive to the gel. This might be because the hydrophobic force between TP and MPs weakened with the increase in TP concentration, which reduced the amount of binding between the two. A large amount of TP dispersed between protein molecules hindered the cross-linking between the proteins, contributing to the breakdown of the gel network structure [32]. This fragmentation not only reduced gel strength but also diminished water retention due to more numerous pores. Across all TP groups, changes in gel strength and water retention were consistent. However, in the GAE-treated groups, both the low- and medium-concentration groups displayed a more continuous and intact network structure compared with the oxidized control group. This indicates that appropriate concentrations of GAE could effectively hinder the oxidative destruction of the gel network structure. This protective effect was likely due to GAE’s strong free radical scavenging ability, which could impede the oxidative process and protect the protein side-chain structure, thereby preserving the integrity of the gel network [33].

## 4. Conclusions

The effects of TP/GAE at different concentrations on the gel properties of MPs were investigated by constructing an •OH-MP oxidation system. The results demonstrate that the regulation of oxidative stability and gel properties of MPs by TP and GAE exhibit a concentration-dependent effect. The addition of an appropriate amount of TP/GAE strengthened the network structure of the MP gels, resulting in a denser gel system. Observations of the gel microstructure indicate that higher concentrations of TP/GAE can lead to poor gel quality, characterized by gel fragmentation and porosity. This disrupts the gel network structure and reduces water retention in the gels. Therefore, adding TP/GAE in appropriate concentrations effectively enhanced the stability of gel properties of MPs, whereas excessive amounts seriously compromised these properties. Based on the obtained results, a concentration of 0.2 mg/g protein is recommended as the ideal amount of GAE for use in actual meat processing to regulate the oxidation and gel properties of meat products.

## Figures and Tables

**Figure 1 foods-13-01447-f001:**
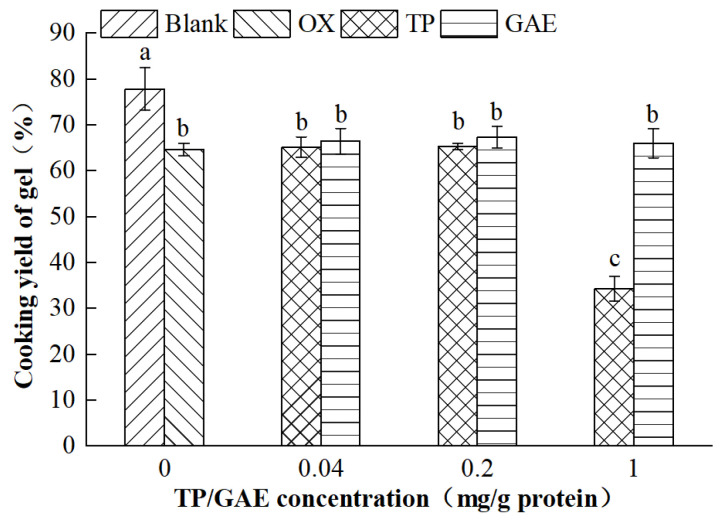
Effects of oxidation and different concentrations of TP/GAE on the cooking yield of MP gels. Note: Different lowercase letters indicate significant differences between different treatment groups (*p* < 0.05). Blank group: no added TP/GAE + PIPES buffer + Trolox solution; OX group: no added TP/GAE + Fenton oxidation system + Trolox solution; TP group: added 0.04–1.0 (mg/g protein) TP + Fenton oxidation system + Trolox solution; GAE group: added 0.04–1.0 (mg/g protein) GAE + Fenton oxidation system + Trolox solution, the same below.

**Figure 2 foods-13-01447-f002:**
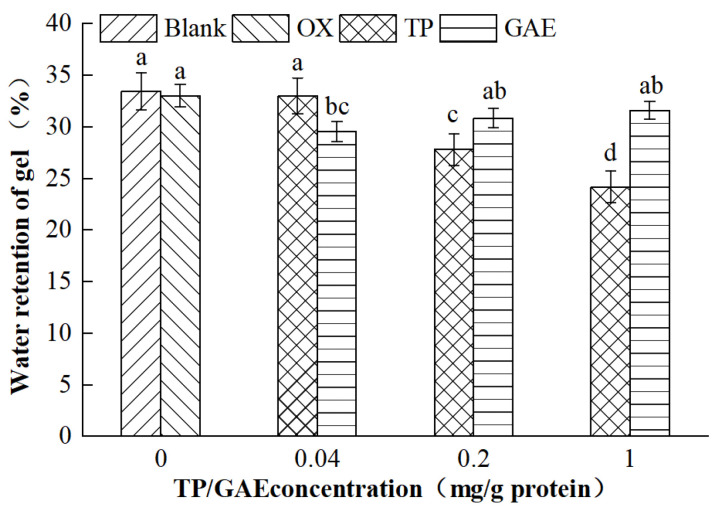
Effects of oxidation and different concentrations of TP/GAE on the water retention of MP gels.

**Figure 3 foods-13-01447-f003:**
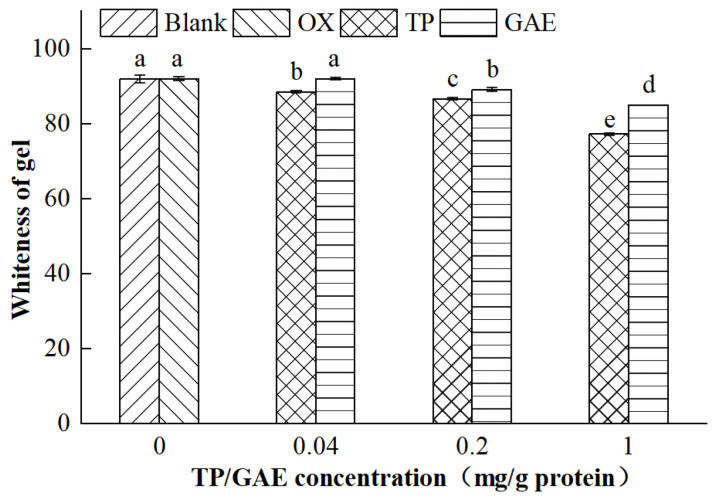
Effects of oxidation and different concentrations of TP/GAE on the whiteness of MP gels.

**Figure 4 foods-13-01447-f004:**
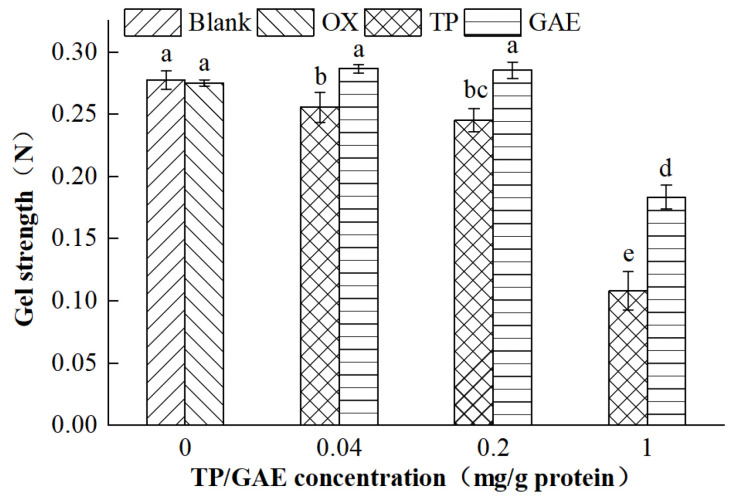
Effects of oxidation and different concentrations of TP/GAE on the strength of MP gels.

**Figure 5 foods-13-01447-f005:**
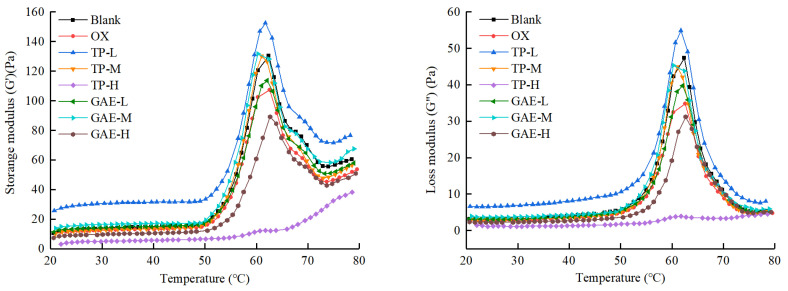
Effects of oxidation and different concentrations of TP/GAE on energy storage modulus (*G*′) and loss modulus (*G*″) of MPs during thermal gelation.

**Figure 6 foods-13-01447-f006:**
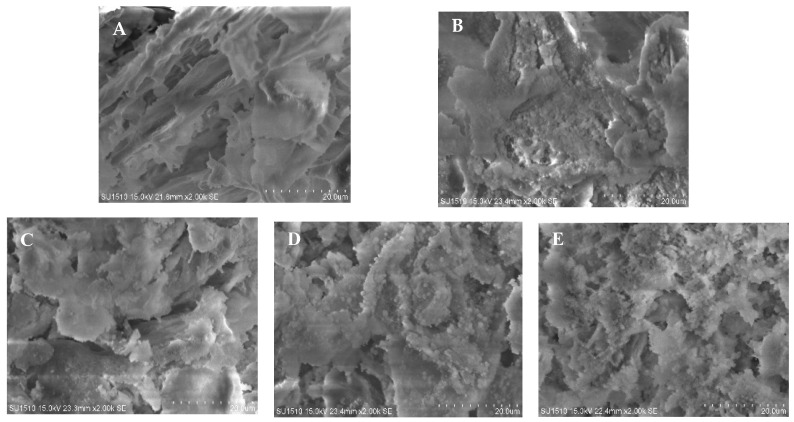
Effects of oxidation and different concentrations of TP/GAE on the microstructure of MP gels. Note: (**A**–**H**) are shown in sequence: OX, TP-L, TP-M, TP-H, GAE-L, GAE-M, GAE-H, magnification 2000.

**Table 1 foods-13-01447-t001:** Experimental samples.

Groups	Concentration (mg/g Protein)	Processing Methods
Blank	0	no added TP/GAE + PIPES buffer + Trolox solution
OX	0	no added TP/GAE + Fenton oxidation system + Trolox solution
TP-L	0.04	added 0.04 (mg/g protein) TP + Fenton oxidation system + Trolox solution
TP-M	0.2	added 0.2 (mg/g protein) TP + Fenton oxidation system + Trolox solution
TP-H	1.0	added 1.0 (mg/g protein) TP + Fenton oxidation system + Trolox solution
GAE-L	0.04	added 0.04 (mg/g protein) GAE + Fenton oxidation system + Trolox solution
GAE-M	0.2	added 0.2 (mg/g protein) GAE + Fenton oxidation system + Trolox solution
GAE-H	1.0	added 1.0 (mg/g protein) GAE + Fenton oxidation system + Trolox solution

**Table 2 foods-13-01447-t002:** Effects of oxidation and TP/GAE addition on texture properties of MP gels.

Groups	Texture Parameter					
	Hardness/g	Springiness	Cohesiveness	Gumminess	Chewiness	Resilience
Blank	121.59 ± 5.10 ^a^	0.92 ± 0.04 ^a^	0.60 ± 0.01 ^a^	72.83 ± 3.76 ^a^	64.08 ± 1.66 ^a^	0.38 ± 0.01 ^a^
OX	72.93 ± 5.88 ^c^	0.81 ± 0.07 ^bc^	0.41 ± 0.04 ^bc^	34.49 ± 4.15 ^bc^	32.83 ± 1.93 ^c^	0.27 ± 0.02 ^b^
TP-L	75.42 ± 8.15 ^c^	0.83 ± 0.08 ^abc^	0.46 ± 0.02 ^bc^	26.69 ± 3.44 ^de^	33.52 ± 1.88 ^c^	0.24 ± 0.02 ^b^
TP-M	66.25 ± 11.36 ^c^	0.73 ± 0.05 ^cd^	0.38 ± 0.03 ^c^	15.74 ± 2.87 ^f^	25.32 ± 3.26 ^d^	0.11 ± 0.03 ^c^
TP-H	61.16 ± 5.62 ^c^	0.61 ± 0.04 ^e^	0.23 ± 0.03 ^d^	37.47 ± 1.36 ^b^	10.87 ± 2.37 ^e^	0.08 ± 0.02 ^c^
GAE-L	74.43 ± 5.27 ^c^	0.69 ± 0.02 ^de^	0.41 ± 0.05 ^bc^	22.98 ± 2.78 ^e^	32.21 ± 3.41 ^c^	0.27 ± 0.03 ^b^
GAE-M	103.40 ± 15.66 ^b^	0.86 ± 0.03 ^ab^	0.48 ± 0.04 ^b^	31.09 ± 2.63 ^cd^	40.49 ± 2.93 ^b^	0.28 ± 0.02 ^b^
GAE-H	82.96 ± 8.15 ^c^	0.72 ± 0.02 ^cd^	0.44 ± 0.02 ^bc^	13.69 ± 1.60 ^f^	36.60 ± 1.42 ^bc^	0.27 ± 0.03 ^b^

Note: Data are expressed as mean ± standard deviation of three replicates (*n* = 3). Different lowercase letters in the same column indicate significant differences in the mean values of each index (*p* < 0.05). Blank group: no added TP/GAE + PIPES buffer + Trolox solution; OX group: no added TP/GAE + Fenton oxidation system + Trolox solution; TP group: added 0.04–1.0 (mg/g protein) TP + Fenton oxidation system + Trolox solution; GAE group: added 0.04–1.0 (mg/g protein) GAE + Fenton oxidation system + Trolox solution.

## Data Availability

The original contributions presented in the study are included in the article, further inquiries can be directed to the corresponding author.

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
