# Peer review of "Effects of Gnaphalium affine Extract on the Gel Properties of •OH-Induced Oxidation of Myofibrillar Proteins"

_foods, 2024, doi:10.3390/foods13101447_

Round 1

Reviewer 1 Report

Comments and Suggestions for Authors

The manuscript is prepared with great attention to details and is written smoothly what makes the text clear and enjoyable to read. However there are some major issue that must be addressed by authors before further processing:

1. The main issue is the novelty of the study. There are a plenty of different studies on the effect of polyphenols on the structure/strenght etc of proteins gels. What is new about the study? Please notice that authors have already published very similar study. What is new about this one? Please clarify and highlight this in abstract and introduction section.

2. Why the authors chose the extract of Gnaphalium affine because it is not highlighted in the study. Authors stated that they want to use the extracts for actual processing of meat. Such extracts should be suitable to use in meat processing. If Gnaphalium affine extract complement the sensorial properties of meat? or is neutral for taste/flavor of meat? What is special about this certain extract to focus on it? Please clarify

And some minor issues as well

3. Line 26-27 there is no data on the subject in the manuscript it is better to reformulate the sentence for " based on the obtained results is reccomended to..." please reformulate here in abstract and in conclusions

4. Line 50 " Plant polyphenols are the most effective natural antioxidants in recent years" It's not clear what authors meant to communicate here. Please reformulate the sentence

5. Lines 57-62 Please add here an explenation why Gnaphalium affine was chosen to be deeply analyzed in the study and as a additive in meat processing itself. 

6. Line 131 Please reformulate the sentence

7. Figures and tables should be self explenatory. Figures 1-4. Please add description for blank, ox as well as for TP and GAE groups in the footnotes . 

8. Line 242 The statment is of a great simplification because the higher amount of fat in meat is also responsible for its lightness not necessairly for the freshness. Please reformulate the sentence.

Author Response

Reviewer #1: 

  1.  Response to comment:(The main issue is the novelty of the study. There are a plenty of different studies on the effect of polyphenols on the structure/strenght etc of proteins gels. What is new about the study? Please notice that authors have already published very similar study. What is new about this one? Please clarify and highlight this in abstract and introduction section.)

Response: Gnaphalium affine extract (GAE) from a kind of traditional wild vegetable featured in the Qingming festival held in China, which rich in polyphenols and flavonoids, and exert the antioxidant effects, and the extract has a strong ability to scavenge free radicals, especially for •OH. Our previous research evaluated the effects of Gnaphalium affine extract (GAE) and sodium pyrophosphate (PP) on the oxidative stability of myofibrillar proteins (MPs) (Chang, H.; Shi, Y.; Xiong, J.; Bo, Z.; Dai, Y. Effects of Gnaphalium affine extract (GAE) and sodium pyrophosphate (PP) on •OH-induced oxidation and the structure of myofibrillar proteins. Int. J. Food Sci. Technol. 2024, 59, 2042-2055. ). Myofibrillar proteins (MPs) are the primary muscle proteins, which play a crucial role in modulating the three-dimensional network of heat-induced gels in meat products. The effects of GAE on the functional properties of MPs in oxidation systems remain less well-known, and the regulation of the gel properties of MPs with GAE in a simulated Fenton oxidation system has not been reported to date. Therefore, further studies are needed to determine whether GAE can mediate the gel properties of MPs under oxidizing conditions. Based on this, the aim of this study was to investigate the effect of GAE on the gel properties of porcine MPs in a simulated Fenton oxidation system. Thank you for your comments.

  1.  Response to comment:(Why the authors chose the extract of Gnaphalium affine because it is not highlighted in the study. Authors stated that they want to use the extracts for actual processing of meat. Such extracts should be suitable to use in meat processing. If Gnaphalium affine extract complement the sensorial properties of meat? or is neutral for taste/flavor of meat? What is special about this certain extract to focus on it? Please clarify)

Response: Gaphalium affine extract (GAE) is rich in polyphenols and flavonoids, belongs to plant polyphenols, which exert a multitude of physiological effects, including antibacterial, anti-inflammatory, and antioxidant effects, and the extract has a strong ability to scavenge free radicals, especially •OH. The application of GAE in meat processing based on that GAE can inhibit protein oxidation instead of synthetic antioxidants. Whether GAE affects functional properties of myofibrillar proteins under oxidation condition needs further exploration. Therefore, investigating the effect of GAE on the gel properties of MPs based on its ability to resist the oxidation of MPs is of great significance for meat processing. 

  1. Response to comment:(Line 26-27 there is no data on the subject in the manuscript it is better to reformulate the sentence for " based on the obtained results is reccomended to..." please reformulate here in abstract and in conclusions)

Response: The sentence in the abstract and conclusions section has been reformulated according to the reviewers' comments, we appreciate your advice very much.

  1. Response to comment:(Line 50 " Plant polyphenols are the most effective natural antioxidants in recent years" It's not clear what authors meant to communicate here. Please reformulate the sentence)

Response: We understand the reviewer’s suggestions, amendments and additions were made according to the comments of the reviewers. Thank you very much for your constructive comments.

  1. Response to comment:(Lines 57-62 Please add here an explenation why Gnaphalium affine was chosen to be deeply analyzed in the study and as a additive in meat processing itself. )

Response: The answer is the same as the first and second questions above (Response to the first and second comment), and we made the revision in the light of the reviewer's comments.

  1. Response to comment:(Line 131 Please reformulate the sentence)

Response: We made some supplements according to the comments of Reviewer’s. Thanks very much for your comments.

  1. Response to comment:(Figures and tables should be self explenatory. Figures 1-4. Please add description for blank, ox as well as for TP and GAE groups in the footnotes)

Response: Relevant additions have been made.

  1. Response to comment:(Line 242 The statment is of a great simplification because the higher amount of fat in meat is also responsible for its lightness not necessairly for the freshness. Please reformulate the sentence.)

Response: We strongly agree with the reviewer that the higher the fat content, the lighter the meat, however, the effects of phenols on water retention of MP gels were analyzed in this section, rather than the effect of fat content on weight of meat.

We appreciate for your good comments greatly, special thanks to you.

Reviewer 2 Report

Comments and Suggestions for Authors

The author's aimed at evaluating the antioxidant activity of a Gnaphalium affine-derived extract (GAE) oxidation (H2O2/FeCl3/ascorbic acid)-prone (Fenton-like) protein (L. dorsi, myofibrillar proteins) system. GAE [at 0.04 (L), 0.2 (M), 1.0 (H) mg.g-1 protein] antioxidant capacity was compared to a tea polyphenol-rich extract (TP; same three concentration levels) in oxidized and non-oxidized protein-rich systems. GAE (M>L,H)>TP performed better (textural properties) in the oxidated system with no apparent differential effect in non-oxidized one. Apparently stable protein systems were seen at L-M but not H concentrations of GAE/PT. Authors concluded that 0.2 GAE is ideal to prevent oxidation while maintaining the original physicochemical properties of cooked pork myofibrillar proteins. Although the experimental design and its execution is good, authors may consider the following to improve their manuscript´s scientific soundness and uniqueness:

General. A) English syntax and grammar should be reviewed by a native English speaker. B) Figures and tables should be located closest to their quote within the text.

Abstract. It should be described in a more objective way (descriptive) considering only what is reported (do not make inductions), highlighting; A) The specific differences of TP vs. GAE (regardless of its concentration) in the most relevant evaluated parameters and B) The most relevant effects of the intermediate concentration of GAE.

Introduction. Quite long. Certain arguments could be used in discussion section (e.g. third paragraph).

Methods. A) The authors should include the phenolic concentration (total phenols, EAG.mg-1)/antioxidant capacity (TEAC, trolox equivalents) of each of the three concentrations of TP and GAE respectively. B) Include data on the antioxidant phytochemical profile of GAE (see (doi): 10.3390/molecules18077751, 10.19386/j.cnki.jxnyxb.2019.10.11, 10.3390/molecules18078298). C) Anticipating that many dependent variables were measured for the same experimental treatments, the authors should consider the use of grouping techniques (e.g. HCA, PLS-DA, PCA) to deduce the most convenient treatment(s) and their individual effects on the parameters. analyzed.

Figures & Tables. A) The sharpness and size of all figures and tables´ format should be improved in accordance with the journal's guidelines. B) Locate title (up) and footnotes (down), the latter detailed as much as possible to be able to understand them without having to read the main text.

References. A) Authors should take a second look on the correct format of each reference according to Foods´ guidelines.

Comments on the Quality of English Language

Moderate editing is needed

Author Response

Reviewer #2:

  1.  Response to comment:( A) English syntax and grammar should be reviewed by a native English speaker. B) Figures and tables should be located closest to their quote within the text)

Response: A) English syntax and grammar have been improved and reviewed by a native English speaker in revisions, and the revisions are marked in red in the paper.  B) Figures and tables have been located closest to their quote within the text.

  1. Response to comment:(It should be described in a more objective way (descriptive) considering only what is reported (do not make inductions), highlighting; A) The specific differences of TP vs. GAE (regardless of its concentration) in the most relevant evaluated parameters and B) The most relevant effects of the intermediate concentration of GAE.)

Response: Part of statements in this section has been reformulated according to the reviewers' comments, we appreciate your advice very much.

  1. Response to comment:(Quite long. Certain arguments could be used in discussion section (e.g. third paragraph).)

Response: Thanks for your comments very much. In view of the fist reviewers' comments, this part has been corrected and added for the explanation and clarify the purpose of plant polyphenols (Gnaphalium affine extract) was chosen in this study.  

  1. Response to comment:(A) The authors should include the phenolic concentration (total phenols, EAG.mg-1)/antioxidant capacity (TEAC, trolox equivalents) of each of the three concentrations of TP and GAE respectively. B) Include data on the antioxidant phytochemical profile of GAE (see (doi): 10.3390/molecules18077751, 10.19386/j.cnki.jxnyxb.2019.10.11, 10.3390/molecules18078298). C) Anticipating that many dependent variables were measured for the same experimental treatments, the authors should consider the use of grouping techniques (e.g. HCA, PLS-DA, PCA) to deduce the most convenient treatment(s) and their individual effects on the parameters. analyzed.)

Response: We understand the reviewer’s suggestions. Because the content of total flavonoids and total phenols in Gaphalium affine extract, as well as the free radical scavenging capacity (antioxidant capacity) of GAE have been published in our previous research (Chang, H.; Shi, Y.; Xiong, J.; Bo, Z.; Dai, Y. Effects of Gnaphalium affine extract (GAE) and sodium pyrophosphate (PP) on •OH-induced oxidation and the structure of myofibrillar proteins. Int. J. Food Sci. Technol. 2024, 59, 2042-2055. ). Therefore, we did not provide those results. In addition, it is worth mentioning that the reviewer has provided us with valuable learning materials and analytical methods, which we will refer to and apply in future research. We appreciate for your good comments greatly.

  1. Response to comment:(Figures & Tables. A) The sharpness and size of all figures and tables´ format should be improved in accordance with the journal's guidelines. B) Locate title (up) and footnotes (down), the latter detailed as much as possible to be able to understand them without having to read the main text.)

Response: The titles and footnotes of all Figures & Tables were all checked, corrected and added according to the reviewers' comments and the journal's guidelines.

  1. Response to comment:( A) Authors should take a second look on the correct format of each reference according to Foods´ guidelines.)

Response: The format of all references were checked according to Foods´guidelines. 

We are deeply grateful for your comments.

Round 2

Reviewer 2 Report

Comments and Suggestions for Authors

Thank you for accepting my partial comments and suggestions. The manuscript improved substantially

Comments on the Quality of English Language

Minor editing is needed